# Green Extraction Techniques of Phytochemicals from *Hedera helix* L. and In Vitro Characterization of the Extracts

**DOI:** 10.3390/plants12223908

**Published:** 2023-11-20

**Authors:** Adina I. Gavrila, Christina M. Zalaru, Rodica Tatia, Ana-Maria Seciu-Grama, Cristina L. Negrea, Ioan Calinescu, Petre Chipurici, Adrian Trifan, Ioana Popa

**Affiliations:** 1Faculty of Chemical Engineering and Biotechnologies, National University of Science and Technology Politehnica Bucharest, 011061 Bucharest, Romania; adinagav@yahoo.com (A.I.G.); negrea_cristina2001@yahoo.com (C.L.N.); ioan.calinescu@gmail.com (I.C.); petre.chipurici@gmail.com (P.C.); adriantrifan2000@yahoo.com (A.T.); 2Department of Organic Chemistry, Biochemistry and Catalysis, Faculty of Chemistry, University of Bucharest, 050663 Bucharest, Romania; chmzalaru@gmail.com; 3Department of Cellular and Molecular Biology, National Institute of Research and Development for Biological Sciences, 060031 Bucharest, Romania; rodica.tatia@gmail.com (R.T.); ana.seciu@yahoo.com (A.-M.S.-G.)

**Keywords:** microwave, ultrasound, saponins, carbohydrates, polyphenols, antioxidant activity, in vitro cytotoxicity

## Abstract

*Hedera helix* L. contains phytochemicals with good biological properties which are beneficial to human health and can be used to protect plants against different diseases. The aim of this research was to find the most suitable extraction method and the most favorable parameters for the extraction of different bioactive compounds from ivy leaves. Different extraction methods, namely microwave-assisted extraction (MAE), ultrasound-assisted extraction (UAE), and conventional heating extraction (CHE), were used. The most suitable method for the extraction of saponins is MAE with an extraction efficiency of 58%, while for carbohydrates and polyphenols, the best results were achieved via UAE with an extraction efficiency of 61.7% and 63.5%, respectively. The antioxidant activity (AA) of the extracts was also determined. The highest AA was obtained via UAE (368.98 ± 9.01 µmol TR/gDM). Better results were achieved at 50 °C for 10 min of extraction, using 80% ethanol in water as solvent. In order to evaluate their in vitro cytotoxicity, the extracts richest in bioactive compounds were tested on NCTC fibroblasts. Their influence on the DNA content of RAW 264.7 murine macrophages was also tested. Until 200 µg/mL, the extracts obtained via UAE and MAE were cytocompatible with NCTC fibroblasts at 48 h of treatment. Summarizing the above, both MAE and UAE can be employed as green and efficient methods for producing extracts rich in bioactive compounds, exhibiting strong antioxidant properties and good noncytotoxic activity.

## 1. Introduction

Common ivy (*Hedera helix* L.) is a medicinal plant which belongs to Araliaceae family and is native to Asia, North America, and Central, Western, and Southern Europe [1]. Although common ivy is inedible for both humans and animals, its leaves are rich in bioactive compounds with beneficial effects on the human body. In addition, botanical extracts rich in such phytochemicals can be used as biostimulants in horticulture or agriculture fields [2]. Regarding ivy leaves, saponins are the most important among these compounds. The structure of saponins implies the presence of at least one glycosidic bond at the carbon atom C3 between aglycone and a sugar moiety. The hydrolysis of saponins leads to an aglycone, usually named sapogenin, and a sugar moiety. *Hedera helix* mainly contains triterpenoid glycosides. The triterpenoids consist of three monoterpenes of 10 carbon atoms each, which are distributed as 6 isoprene molecules [3]. Among triterpene saponins, hederacoside C, hederagenin, and α-hederin are predominant in *Hedera helix* [4]. *Hedera helix* is also rich in polyphenols. These compounds contain one or more aromatic rings to which one or more hydroxylic groups are attached. In plants, they are usually linked by sugar moieties, being rarely found in their free form [5]. The polyphenols found in *Hedera helix* are caffeic and chlorogenic acids, kaempferol, rutin, and quercetin [6]. In *Hedera helix* leaves, branched-chain monosaccharides, such as hamamelose and apiose, were also found, along with others sugars such as glucose, fructose, sucrose, and raffinose [7].

These bioactive compounds found in *Hedera helix* leaves provide to the extracts immunological [8], cytotoxic [9], antibacterial [10], antiviral [11], antifungal [12], anti-inflammatory [13], antiparasitic [14], antimutagenic [15], and antioxidant [16] properties.

The first step to use these bioactive compounds in various industries, such as pharmaceutical, food, agricultural, cosmetic, etc., involves their extraction from *Hedera helix* leaves. The most common methods, such as heat reflux [17], maceration [18], or Soxhlet [19] extractions, can imply long extraction times, high temperatures, excessive amounts of solvent (sometimes toxic solvents), and substantial energy consumption, resulting in low efficiency and selectivity. Among non-conventional extractions which can overcome these drawbacks are microwave- and ultrasound-assisted extractions. The microwave-assisted extraction (MAE) process consists of heating both the solvent and plant tissue, thus improving the kinetics of extraction. During the microwave heating, the plant material is exposed to microwave energy, creating a temperature gradient that leads to a decrease in the chemical potential of the plant cell which will result in the cell wall rupture and lead, therefore, to an easy release of the targeted compounds [20,21,22,23]. Ultrasound-assisted extraction (UAE) is based on the propagation of acoustic waves in liquids that produces the cavitation phenomenon. It is described by the generation of microbubbles or cavities, which collapse when they reach a maximum size. The presence of a solid material near the collapsing bubbles can produce an asymmetric collapse. Thus, the cavitation can promote rupture of the cell wall and lead, therefore, to an increase in the mass transfer rate between solvent and plant material, while also increasing the permeability of cell walls [24,25,26,27].

The evaluation and screening of the most suitable extraction method and the most favorable parameters for extracting various bioactive compounds from *Hedera helix* leaves are reported in this study. To the best of our knowledge, a comparison of these phytochemicals’ extraction methods has not been reported yet, and the microwave-assisted extraction of different bioactive compounds from *Hedera helix* leaves has been employed, for the first time, in this work. There is only one study which uses microwaves, but the researchers applied a microwave pre-treatment of the *Caulis hederae sinensis* leaves, the extraction of saponins being performed with supercritical fluids [28]. The aim of this study was to evaluate the *Hedera helix* leaves extracts as potential sources of bioactive compounds, such as saponins, carbohydrates, and polyphenols, with therapeutic use. The antioxidant activity of the extracts is also reported. Finally, an in vitro analysis of the extracts richest in bioactive compounds was conducted in order to evaluate their cytotoxicity on NCTC normal fibroblast cells and the influence on the DNA content of RAW 264.7 murine macrophages.

## 2. Results and Discussion

Different methods of extracting phytochemical compounds from plants and herbs lead to different extraction yields and efficiencies. An important factor to consider in the manufacture of bio-based products is the increase in the bioactive compound content. In the scaling-up processes, low energy consumption and economical cost must be considered. This means that a low specific energy (consumed energy per amount of compound) is required, which can be accomplished with a high extraction yield. According to the literature [25,29], among other parameters, extraction time, temperature, and ethanol concentration in water can affect the content of extracted compounds. The aim of this work was to study the effect of these three parameters on different extraction methods of *Hedera helix* leaves and to compare the phytochemical extraction content and antioxidant activity (AA) for the following extraction methods: conventional heating extraction (CHE), UAE, and MAE. The non-conventional techniques (MAE and UAE) used in this work present several benefits such as good control of the heating process, shorter extraction time, better extraction efficiency and selectivity, and reduced levels of required energy.

### 2.1. Multiple Successive Extractions

In order to assess the efficiency of bioactive compound extraction, information about the total amount of extractable phytochemicals in Hedera helix leaves is required. Thus, discontinuous multiple successive extractions were performed using the best extraction conditions obtained for each extraction method. The phytochemical content—total saponin content (TSC), total carbohydrate content (TCC), total phenolic content (TPC)—obtained through multiple successive extractions, for each method, is shown in Figure 1.

The total antioxidant potential of the samples was determined using the cupric reducing antioxidant power (CUPRAC) colorimetric assay. Cu^2^⁺ reduction is a common method for assessing electron donation activity and is an important mechanism of antioxidants. Thus, an investigation into the capacity of Hedera helix extracts to reduce Cu(II) and evaluate their electron-donating abilities was conducted. High values of AA suggest a high reducing activity. Using copper(II)-neocuproine reagent as the chromogenic oxidant, the CUPRAC assay is based on the reduction of Cu(II) to Cu(I) by antioxidants found in plant extracts. Due to their high concentration of saponins, polyphenols, and carbohydrates, acting as electron donors, the extracts may have an antioxidant mechanism that contributes to their cupric reducing ability [30,31]. The AA of the extracts obtained by multiple successive extractions, for each method, is also shown in Figure 1. The highest AA values were obtained for UAE (368.98 ± 9.01 µmol TR/g DM), followed by MAE (305.70 ± 2.36 µmol TR/g DM).

In Figure 1, it can be noticed that the maximum amount of phytochemical compounds was achieved via non-conventional methods (e.g., TSCmax = 133.65 mg DE/gDM and TPCmax = 45.98 mg GAE/gDM via MAE; and TCCmax = 363 mg GE/gDM via UAE). For all bioactive compounds, more than 73% are extracted in the first two extractions, for all methods used. The TSC of first extract varied between 64.5 and 77.64 mg DE/gDM, and the TCC ranged from 298.78 to 363 mg GE/gDM, while the TPC ranged from 27.66 to 45.98 mg GAE/gDM.

The most efficient extraction methods are the non-conventional ones (for saponins, MAE and UAE efficiencies are higher than 99%; for carbohydrates, MAE and UAE efficiencies are higher than 91%; and for polyphenols, MAE and UAE efficiencies are higher than 87%).

When multiple extractions from the same raw material are involved, after each extraction, the plant (although initially dried) will swell up and saturate with the solvent despite centrifugation of the extraction mixture. Thus, a higher number of extraction cycles may increase the plant material-to-solvent ratio if the solvent volume remains constant for each extraction. This could affect the extraction rate and lead to a waste of solvent without an improvement in the content of the extracted phytocompounds. To investigate the influence of various parameters, the extraction efficiencies were compared with the maximum extractable content obtained through the multiple successive extractions.

### 2.2. The Influence of Extraction Time on the Bioactive Compound Content

The extraction time is an important parameter that impacts the content of extracted products. The amount of extracted bioactive phytochemicals increases over time until an equilibrium between the solubilized compounds by the solvent and the un-extracted constituents from the plant cells is reached [32]. The efficiencies of emerging extraction techniques (UAE and MAE) were compared with the conventional method. Different extraction times (1, 5, 10, and 20 min) were employed for CHE, UAE, and MAE. Other parameters, such as temperature (50 °C), solvent concentration (80% aqueous ethanol), and plant material-to-solvent ratio (1/20 *w*/*v*) were maintained constant. The influence of extraction time on the TSC, TCC, TPC, and AA was evaluated. The effect of extraction time on the total content of bioactive compounds and antioxidant activity is shown in Figure 2.

The results presented in Figure 2 reveal that for CHE, UAE, and MAE methods, the total content of bioactive compounds increases with increasing the extraction time up to 10 min, after which it decreases. The significant influence of extraction time on the total content of bioactive compounds and AA was confirmed via ANOVA and Duncan’s post hoc *t*-tests. Considering the best extraction time of 10 min, the following findings emerge:➢Saponins (Figure 2a): TSC increases until 10 min, and after that, by increasing the extraction time, a significant (*p* < 0.05) decrease in extraction yield for all methods is observed. The highest TSC (77.6 ± 1.7 mg DE/gDM) was achieved via MAE, followed by UAE (74.5 ± 1.5 mg DE/gDM) and CHE (64.5 ± 0.6 mg DE/gDM).➢Carbohydrates (Figure 2b): TCC increases significantly until 20 min (*p* < 0.05). UAE led to the highest TCC (224.2 ± 8.5 mg GE/gDM), followed by MAE (200.2 ± 4.0 mg GE/gDM) and CHE (184.0 ± 6.3 mg GE/gDM).➢Polyphenols (Figure 2c): TPC significantly increases until 10 min (*p* < 0.05) with no significant differences observed by extending the extraction time to 20 min. UAE led to the highest TPC (28.3 ± 1.2 mg GAE/gDM), followed by MAE (30.0 ± 0.7 mg GAE/gDM) and CHE (26.1 ± 1.3 mg GAE/gDM).➢Antioxidant activity (Figure 2d): AA is correlated with the phytocompound content, with the highest activity being achieved at 10 min. Extending the extraction time to 20 min, the AA increase was not significant. UAE led to the highest AA (368.9 ± 9.0 mmol TR/gDM), followed by MAE (305.7 ± 2.4 mmol TR/gDM) and CHE (279.0 ± 5.2 mmol TR/gDM).

The maximum TPC obtained through non-conventional methods is comparable to that obtained via maceration in 12 h (28.3 mg GAE/g) as reported by Zaiter et al. [18]. However, in this study by MAE and UAE, the content was achieved in a much shorter time (10 min).

According to Figure 2, the extraction yield increasing is not significant or tends to decrease within 10 and 20 min. Moreover, the extracted compounds might be susceptible to degradation due to a prolonged extraction time. Thus, the best extraction time is considered to be 10 min, when the highest amount of bioactive compounds was obtained.

### 2.3. The Influence of Temperature on the Bioactive Compound Content

The extraction temperature is another important factor, which may affect the solid–liquid extraction process. Due to limitations of the non-conventional methods, 40 and 50 °C were chosen to evaluate the influence of temperature on the content of bioactive compounds. Below 40 °C, the microwaves are not effective, and for temperatures exceeding 50 °C, some limitations can occur for UAE because it is known that the effect of ultrasounds is better at lower temperatures [33]. By increasing the temperature, bubbles can be formed in higher quantity and their collapse is less violent (due to the cushioning effect of vapors in bubble). Thus, the mass transfer enrichment caused by cavitation is reduced. Moreover, excessively high temperatures may alter the molecular structure of phytochemicals, leading to their destruction and implicit reduction in the extraction yield. Saponins, carbohydrates, and polyphenols, along with AA, were quantified for these two temperatures and are presented in Figure 3.

As shown in Figure 3, the highest amount of all bioactive compounds was extracted at a temperature of 50 °C. Increasing the temperature leads to an increase in solubility of bioactive compounds and enhances the mass transfer rate between solvent and vegetal material matrix. A high temperature can determine rapid ruptures of the cell wall and promote the facile release of phytocompounds. However, a long extraction time and a high temperature may lead to the oxidation of bioactive compounds or may modify the conformation of extracted compounds. Moreover, in the case of thermolabile phytocompounds, maintaining the temperature at low values avoids phytochemical degradation [21].

The extraction efficiency of the studied phytochemicals is higher than 55% for the non-conventional methods. A significant influence of the extraction temperature on the total bioactive compound content was shown via an ANOVA test and Duncan’s post hoc *t*-tests (see Figure 3). The MAE technique led to the highest TSC (Figure 3a) with an efficiency of 58% (*p* > 0.05). Using the UAE method, the highest values for the TCC (efficiency of 61.7%, Figure 3b), TPC (efficiency of 65.3%, Figure 3c), and AA (Figure 3d) were achieved. These results are better than those obtained via conventional methods reported in the literature. Tatia et al. achieved values of 30.02 mg DE/g and 199.27 mmol TR/g for TSC and AA, respectively, by heat reflux extraction [17], which are lower than those obtained in this study via UAE and MAE.

### 2.4. The Influence of Solvent Concentration on the Bioactive Compound Content

Another important parameter for phytocompound extraction is the solvent concentration. In order to select the proper solvent for the extraction process, various properties including viscosity, vapor pressure, surface tension, and solubility of compounds must be considered [34]. For example, some solvents remain transparent to microwaves. This means that microwave energy can only be absorbed by dielectric solvents. The higher the dissipation factor of the solvent, the better its microwave absorption [35]. Regarding ultrasounds, the most important properties are the viscosity, vapor pressure, and surface tension of solvent. High surface tension and low viscosity result in increased molecular interactions in the solvent, causing cavitation to be initiated at a higher energy threshold. Cavity bubbles are more violently collapsed in low vapor pressure solvents than in those with high vapor pressure [36].

One solvent that meets these characteristics and can be effectively used for both UAE and MAE is ethanol. Moreover, ethanol is a green solvent widely used for the extraction of bioactive compounds from different plants due to its low toxicity. To improve the extraction efficiency, ethanol is commonly used at different concentrations in water. For this study, different concentrations were used. The influence of solvent concentration on the phytoconstituent content is shown in Figure 4.

The total content of bioactive compounds increases until an ethanol concentration of 80% is reached. The significant influence of ethanol concentration on the extraction yield was confirmed via ANOVA test and Duncan’s post hoc *t*-tests (Figure 4). A further increase in ethanol concentration does not enhance the yield of the target compounds.

As depicted in Figure 4a, the highest TSC is achieved with an ethanol concentration of 80% for MAE. However, there are small differences between MAE and UAE, the recovery efficiencies being 58% and 55.7%, respectively. The extraction of saponins with 96% ethanol resulted in a decrease in the saponin content. This behavior can be explained by the polarity similarities between 80% ethanol and saponins, thus enhancing the extraction of these compounds [37].

The carbohydrate content increases with the ethanol concentration up to 80% but it decreases drastically at a concentration of 96% (see Figure 4b). This can be due to the lower solubility of carbohydrates in ethanol compared to aqueous ethanolic solutions. It can be noticed, in Figure 4b, that the carbohydrate content for 96% ethanol is lower than that extracted in distilled water. When comparing the extraction methods, UAE yielded the highest amount of carbohydrates with a 61.7% efficiency, followed by MAE with a 55.1% efficiency.

Polyphenols exhibit a similar behavior to carbohydrates and saponins: the highest TPC was obtained at an 80% ethanol concentration. The best results were achieved via UAE (65.3% efficiency), followed by MAE with 61.5% efficiency (see Figure 4c).

The AA of the extracts performed in 80% ethanol concentration was significantly higher than that of the extracts carried out in distilled water or with 40 and 96% ethanol concentration (see Figure 4d). As shown in Figure 4a–c, the TSC, TCC, and TPC for the ethanol solutions ranging between 40 and 80% concentration were significantly higher than those obtained with 96% ethanol and distilled water. UAE yielded the highest AA, which is in accordance with the highest content of polyphenols, carbohydrates, and saponins. Consequently, the high content of bioactive compounds in the extracts performed in 80% ethanol led to a significant increase in AA. This was confirmed via ANOVA and Duncan’s post hoc *t*-tests. Upon analysis, it can be concluded that, for the extraction of phytoconstituents from *Hedera helix* leaves using different extraction techniques, the suitable ethanol concentration is 80%.

### 2.5. Principal Component Analysis

Principal Component Analysis (PCA) was performed in order to study possible correlations between the TSC, TCC, TPC, AA and the extraction methods (CHE, UAE, and MAE) of these compounds from *Hedera helix* leaves. A common method for assessing the connection between phytochemicals and antioxidant activity in plants is correlation analysis. For the multivariate analysis, the total phytochemical content and antioxidant capacity of the extracts obtained by different extraction techniques were determined. The bioactive compounds and antioxidant activity in the plane formed by the first two axes (PCs) explain 96.11% (eigenvalues > 1) of the total variation, as well as 91.70% on the first axis and 4.41% on the second one. The coordinates of variables (factor loadings) on the factor-plane PC1-PC2 with significant levels marked in bold are shown in Appendix A. The projections of cases (factor scores) on the factor-plane PC1-PC2 and the description of each method are summarized in Appendix A.

The PCA bi-plot and the correlation matrix are shown in Figure 5 and Appendix A, respectively. The significant values of correlation coefficients (*r*) are highlighted in bold at a significance level α = 0.05 (two-tailed test).

From the information of the factor loadings and scores presented in Appendix A and Figure 5, it is possible to correlate the extraction method and extraction conditions with the total bioactive compound content and antioxidant capacity. Thus, the following observations were made:➢TCC, TPC, and AA exhibit a positive influence on the PC1, and TSC shows a positive influence on PC2. Moreover, there is a high correlation between TCC, TPC, and AA. AA is highly positively correlated with TCC (*r* = 0.928) and TPC (*r* = 0.922), and positively correlated with TSC (*r* = 0.867)—the angles between vectors are lower than 90 degrees;➢Based on the PCA plot, two main groups of extraction samples were distinguished: group 1–5 (green ellipsoid) and group 6–9 (red circle). The extracts obtained via methods 1–5 (highlighted by green ellipsoid) had the higher content of bioactive compounds and the highest antioxidant activity compared with the extracts obtained via methods 6–9 (highlighted using red circle—discrimination on PC1).

### 2.6. Energy Considerations

During all experiments, the ultrasound and microwave powers were recorded. The power input for the heating plate, utilized for the conventional extraction, was measured using a Wattmeter. These recorded powers were used to determine the total energy introduced into the systems. Further, the specific energy was calculated using the following equation:(1)$$E_{s}=E_{total}/m_{Phc}\;\left[kJ/g\;of\;phytocompounds\right],$$
where Es is the specific energy [kJ/g of phytocompounds], E_total_ is the total energy introduced into the system [kJ], and m_Phc_ is the total amount of phytocompounds (the sum of saponins, carbohydrates, and polyphenols) obtained via each extraction method [g].

The total and specific energies for each method are presented in Table 1. Energy consumption for CHE ranges between 360 and 378 kJ, while for MAE and UAE, it ranges from 7.1 to 9.2 kJ and from 16.2 to 28 kJ, respectively. Although the lowest energy is consumed for extraction at a temperature of 40 °C, for both UAE and MAE, the lowest specific energy is achieved for the experiments performed at a temperature of 50 °C using an 80% ethanol concentration in water (see Table 1, lines 3 and 4). Under these extraction conditions, the specific energy for CHE is 47 and 25 times higher compared with MAE and UAE, respectively. Regarding the ethanol concentration in water, it can be noticed in Table 1 that the best results are achieved with 80% ethanol. The energy considerations presented in Table 1 reveal that MAE is the greenest method, leading to the lowest specific energy, regardless of extraction conditions, when all phytocompounds (saponins, carbohydrates, and polyphenols) from the same extract are considered. However, in comparison with CHE, both non-conventional methods resulted in significantly lower energy consumption, regardless of extraction conditions. Therefore, in order to obtain extracts rich in bioactive compounds with strong antioxidant properties (such as saponins and polyphenols) both MAE and UAE procedures can be considered.

### 2.7. Biological Avtivity

The data obtained from the in vitro cytotoxicity evaluation using MTT spectrophotometric method quantify the proportion of living cells in NCTC culture after 48 h of treatment with various concentrations of the tested extracts. Cell proliferation of NCTC fibroblasts values after 48 h of treatment, depending on concentration of the sample, are shown Figure 6.

The MTT assay at 48 h of treatment (for concentrations between 10 and 200 µg/mL) indicated that the extracts obtained via UAE and MAE methods with 80% ethanol manifested a high biocompatibility with NCTC fibroblasts, with cell proliferation values significantly higher (*p* < 0.05) than 90%. On the contrary, the extracts obtained via CHE and MAE in 0% ethanol were cytotoxic, generating a low proliferation rate (lower than 56%). Following the obtained results for the interaction of plant extracts with NCTC fibroblasts, the correlation between the tested concentration of the extracts and the cell proliferation rate was observed.

A Live and Dead assay was performed to confirm the noncytotoxic activity of the plant extracts obtained via UAE and MAE in 80% ethanol, against RAW267.4 cell line. The results obtained after 48 h of treatment are shown in Figure 7.

This flow cytometry technique facilitated the highlighting and discrimination of live cells from dead cells, with the analysis showing that the cells cultured in the presence of the tested samples are positive for calcein-AM in a proportion of over 82% (*p* < 0.05). These results indicate a good biocompatibility of the tested samples at 48 h of cultivation, at all concentrations tested.

## 3. Materials and Methods

### 3.1. Materials

Ivy leaves (*Hedera helix* L.) were collected in July 2022 at Hofigal S.A, Romania. The leaves were harvested manually, without stems. To ensure the reproducibility of the extraction methods, leaves from the same lot were used. Information about the ivy leaves used in this study can be obtained from the voucher plant specimen number 407,754, which is archived at the Botanical Garden in Bucharest, Romania. With the help of an air flow-heating oven, the fresh ivy leaves were dried at 60 °C to constant mass. The final humidity was 5.8%. The dried material was powdered and sieved to particles less than 315 μm in size. The dried and milled ivy leaves were dosed and sealed in plastic vessels samples of 25 g. The material was stored at 4–5 °C until it was employed for the extraction of phytochemicals.

The standards used for saponins, carbohydrates, polyphenols, and antioxidant activity determination were diosgenin, glucose, gallic acid, and Trolox, respectively, purchased from Sigma-Aldrich Co, Romania. The solvents ethanol and methanol, vanillin, sulfuric acid, phenol, Folin–Ciocalteu reagent, sodium carbonate, copper chloride, neocuproine, and ammonium acetate were of analytical grade and were purchased from Merck, Germany.

The in vitro experiments were performed using a NCTC (L929 clone) of mouse normal fibroblasts cell line and a RAW 267.4 murine macrophages cell line. Both cell lines were purchased from ECACC (Sigma-Aldrich Co., Bucharest, Romania) together with cell culture media of Minimum Essential Medium (MEM) and Roswell Park Memorial Institute 1640 Medium (RPMI). Fetal bovine serum (FBS) was produced by Biochrom, L-glutamins, and the antibiotics: penicillin, streptomycin, and neomycin were purchased from Sigma Aldrich. The cytotoxicity assay of the extracts on NCTC cells was performed using 3-(4,5-dimethylthiazol-2-yl)-2,5-diphenyltetrazolium bromide assay (MTT), purchased from Sigma Aldrich. For the assessment of cell viability via flow cytometry on RAW 267.4 cells, the LIVE/DEAD Viability Kit (Invitrogen) was used.

### 3.2. Bioactive Compound Extraction Procedure

Three extraction methods are compared in the present paper: CHE, UAE, and MAE. The extractions of bioactive compounds were carried out in a batch system.

All three methods (CHE, UAE, and MAE) were conducted under the following extraction conditions:➢Mixtures of 0, 40, 80, and 96% ethanol in water as solvent;➢Plant material-to-solvent ratio of 1/20 (*w*/*v*);➢Temperatures of 40 and 50 °C;➢Extraction times of 1, 5, 10, and 20 min.

The UAE method was performed in a special jacketed glass reactor using a Vibracell VCX750 (Sonics&Materials, Inc.; Newtown, CT, USA) ultrasonic probe. The extraction temperature was kept at the proposed value by introducing water at convenient temperature into the extraction vessel jacket during all experiments. The sonication was applied in continuous mode, at an amplitude of 40%. The latter was chosen based on our previous work, Gavrila et al. [4].

The MAE method was carried out in a standard Biotage reactor using the microwave applicator Biotage^®^Initiator (Biotage Sweden AB; Uppsala, Sweden). The experiments were conducted at a stirring rate of 900 rpm. The microwave power was between 12 and 27 W depending on the extraction time, temperature, and ethanol concentration. The latter was chosen based on our previous work, Asofiei et al. [38].

The CHE experiments were performed in order to compare the efficiency of non-conventional extraction methods (UAE and MAE) of bioactive compounds from ivy leaves. The experiments under CHE were carried out in the same conditions as for non-conventional techniques using the jacketed glass reactor as for UAE, but without applying the ultrasounds, and using a stirring rate of 900 rpm.

After extraction via each method, the mixture was centrifuged at 2500 rpm for 5 min, at room temperature. The supernatant was collected and freshly analyzed every time. All experiments were performed in triplicate.

### 3.3. Multiple Successive Extraction Procedure

The procedure of discontinuous multiple extraction implies adding a fresh portion of solvent over the same raw material after the mixture is centrifuged. This approach was repeated until the plant material was exhausted (six times), and each fraction collected was freshly analyzed. The discontinuous multiple extraction was performed in triplicate for all three methods.

The extraction efficiency of each phytochemical compound was determined using the following equation:(2)$$E\left(\%\right)=\frac{Total\;amount\;obtained\;by\;each\;method}{Maximum\;extractable\;content}\;\times \;100$$

### 3.4. Phytochemical Content Analysis

Phytochemical contents, such as saponins, carbohydrates, phenolics, and antioxidant activity were quantified using colorimetric methods. For all analyses, the absorbance was measured using a Shimadzu UV mini-1240 UV/Vis Scanning Spectrophotometer, 115 VAC (Shimadzu Deutschland GmbH; Duisburg, Germany). Each analysis was performed in triplicate for all extraction methods.

The total saponin content (TSC) was colorimetrically evaluated using the modified Hiai method [39], described in our previous work [4]. The absorbance was measured at 544 nm. The TSC was expressed as milligrams of diosgenin equivalents per 1 g of dry matter (mg DE/g DM) based on a standard curve corresponding to 40–550 mg/L diosgenin solution.

The total carbohydrate content (TCC) was determined colorimetrically using the method reported by Varkhade et al., with minor modifications [40]. Briefly, 1 mL of diluted extract, 1 mL of 5% phenol solution in water, and 5 mL of 98% sulfuric acid were added to a vial and mixed thoroughly, then left to rest at room temperature for 10 min. Further, the vial was placed in a water bath and maintained at 25–30 °C for 20 min, under constant stirring. The absorbance was measured at 490 nm. The quantification of TCC was presented as milligrams of glucose equivalents per 1 g of dry matter (mg GE/g DM) considering the standard curve which corresponds to 30–280 mg/L glucose solution.

The total phenolic content (TPC) was colorimetrically evaluated. This determination is described in our previous work [41]. The absorbance was measured at 760 nm. The TPC quantification was given as milligrams of gallic acid equivalents per 1 g of dry matter (mg GAE/g DM) based on a standard curve corresponding to 1–5 mg/mL gallic acid solution.

The antioxidant activity (AA) of the extracts was determined using the CUPRAC assay reported by Özyürek et al. [42], with minor modifications. Briefly, 1 mL of 0.01 M cooper (II) chloride aqueous solution, 1 mL of 0.0075 M neocuproine ethanolic solution, 1 mL of ammonium acetate buffer solution, and 1.1 mL of diluted extract were added to a vial and mixed thoroughly. Further, the mixture was kept in the dark, at room temperature, for 30 min. The absorbance was measured at 450 nm. The results were expressed as milligrams of Trolox equivalents per 1 g of dry matter (µmol TE/g DM). A standard curve which corresponds to 0–0.25 mg/mL Trolox solution was used for the AA quantification.

### 3.5. Cytotoxicity Assay

The biocompatibility of the plant extracts was evaluated on a stabilized NCTC mouse fibroblast cell line at 48 h of treatment using the cell viability MTT assay, carried out in accordance with the European standard SR EN ISO 10993-5:2009. The cells were seeded in culture plates with 96 wells at a cell density of 4 × 104 cells/mL in MEM culture medium with 10% fetal bovine serum and 1% antibiotics. They were incubated at 37 °C in a humid atmosphere with 5% CO_2_, for 24 h. The experiment proceeded by putting the cells from the wells in direct contact with the solutions of plant extracts, at concentrations of 10, 50, 100, 200, and 400 µg/mL prepared in culture medium. At 48 h of treatment, the extracts in variable concentrations were replaced with a tetrazolium salt solution (5 mg/mL), and the experimental plate was incubated for 3 h under standard conditions, during which the specific reaction of the MTT assay generated formazan crystals in viable NCTC cells. After 3 h of incubation, the solution was removed from the wells and the formazan crystals were solubilized by adding isopropanol, resulting in a blue-violet coloring of the solutions from the wells. The spectrophotometric measurement of the color intensity of the solutions was carried out using a Berthold Mithras UV-Vis LB940 spectrophotometer (Germany), at a wavelength of 570 nm. The results were calculated as a percentage, based on untreated control cells, whose viability was considered 100%.

### 3.6. Live and Dead Staining

The assessment of cell viability of RAW 264.7 cells cultured for 48 h in the presence of the extracts was carried out via flow cytometry using the LIVE/DEAD Viability Kit, according to the manufacturer’s recommendations (Invitrogen). For the analysis of cell viability, RAW 267.4 macrophages were seeded in 12-well culture plates, at a cell density of 1.5 × 105 cells/mL. The RAW 267.4 cells were cultured in RPMI medium supplemented with 10% fetal bovine serum and 1% antibiotics (penicillin, streptomycin, and neomycin), being maintained at 37 °C, in a humid atmosphere with 5% CO_2_. After 24 h, the cells were treated with samples in concentrations of 10, 50, and 100 μg/mL. After 48 h of cultivation in the presence of the samples, the cells were trypsinized and washed with phosphate-buffered saline (PBS). Further, the cells were labeled with Live and Dead kit through incubation with calcein-AM and ethidium homodimer-1 for 20 min. The cells were then washed, resuspended in PBS, and analyzed using the LSR II flow cytometer (Becton Dickinson) coupled with the FACSDiva software package.

### 3.7. Statistical Analysis

All measurements were conducted in triplicate with the data expressed as mean value ± SD (standard deviation, *n* = 3). All the results obtained at various levels of process factors were evaluated via univariate ANOVA, and a multivariate principal component analysis (PCA) was performed. Duncan’s new multiple comparison test at 95% confidence level was used for statistical analysis of the data, in order to detect the significant statistical differences between the averages of the main constituents of two or more independent groups. To estimate the power of linear correlations between dependent variables, the correlation Pearson coefficient (r) was used [43,44]. Statistical analysis was performed with the help of XLSTAT Version 2019.1 (Addinsoft, New York, NY, USA).

## 4. Conclusions

In this study, the most efficient extraction method of the active principles from ivy leaves in batch mode via CHE, UAE, and MAE was investigated. In order to assess the influence of different factors on the extraction process efficiency, a series of parameters were studied: extraction time, ethanol concentration in the extraction solvent, temperature, and multiple successive extraction. UAE and MAE have been shown to be an attractive alternative to the conventional method (CHE). The results indicated that the best parameters were an extraction time of 10 min, a temperature of 50 °C, and 80% ethanol as extraction solvent. For the extraction of saponins, the highest value of TSC was obtained via MAE (77.64 ± 1.79 mg DE/gDM). As for carbohydrates and polyphenols, the best results were achieved via UAE (TCC = 224.17 ± 8.53 mg GE/gDM and TPC = 30.02 ± 0.75 mg GAE/gDM). PCA was employed to explore potential correlations between TSC, TCC, TPC, AA, and extraction methods (CHE, UAE, and MAE) of the bioactive compounds from *Hedera helix* leaves. A positive correlation between antioxidant activity and TSC (*r* = 0.867), TCC (*r* = 0.928), and TPC (*r* = 0.922) was observed. The in vitro cytotoxicity experiment of ivy extracts assayed on NCTC normal fibroblasts showed a high degree of biocompatibility for the extracts obtained via UAE and MAE in 80% ethanol, within a concentration range of 10 to 200 µg/mL. The results were confirmed via Live and Dead flow cytometry assay performed on RAW 264.7 murine cells. On the contrary, the extract obtained via CHE in 0% ethanol, also tested in vitro on NCTC cells, manifested high cytotoxicity at concentrations higher than 200 µg/mL.

The resulting *Hedera helix* extracts could potentially find applications in the pharmaceutical field as alternative sources of antioxidant, antimicrobial, and anti-inflammatory products. Overall, future studies focused on identification of the compounds and the optimization of bioactive compound extraction should be performed for a comprehensive evaluation of non-conventional extractions and their overall potential.

## Figures and Tables

**Figure 1 plants-12-03908-f001:**
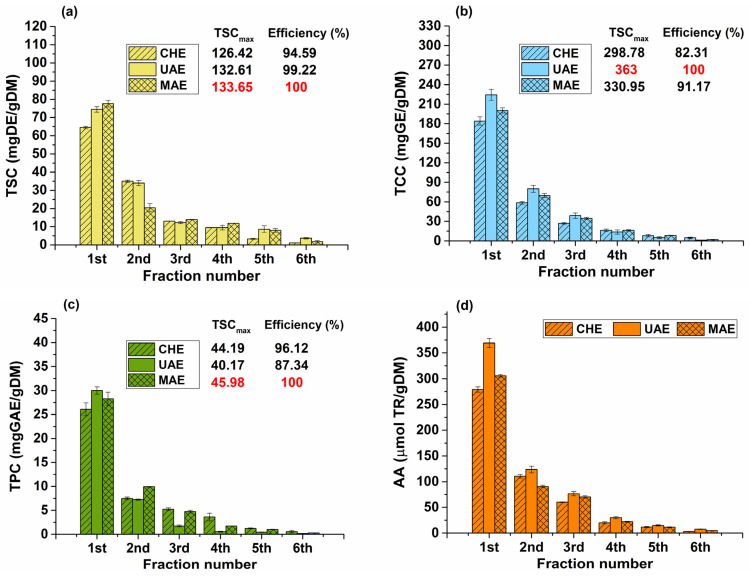
The phytochemical content of the multiple successive extractions: (**a**) total saponin content (TSC), (**b**) total carbohydrate content (TCC), (**c**) total phenolic content (TPC), and (**d**) antioxidant activity (AA). Extraction conditions: 10 min, 80% ethanol in water, 50 °C—conventional heating extraction (CHE), ultrasound-assisted extraction (UAE), microwave-assisted extraction (MAE).

**Figure 2 plants-12-03908-f002:**
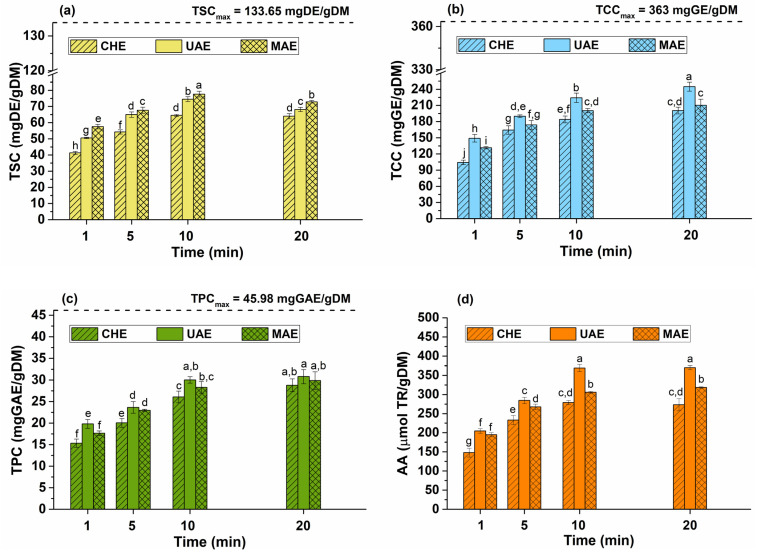
Influence of extraction time on the total content of bioactive compounds: (**a**) TSC, (**b**) TCC, (**c**) TPC, and (**d**) AA—ivy leaves-to-solvent ratio of 1/20 (*w*/*v*), temperature of 50 °C, ethanol concentration in water of 80%, stirring rate of 900 rpm, ultrasound amplitude of 40%. The significant difference between groups (*p* < 0.05, ANOVA and Duncan’s post hoc *t*-tests) are highlighted by different letters (a–j).

**Figure 3 plants-12-03908-f003:**
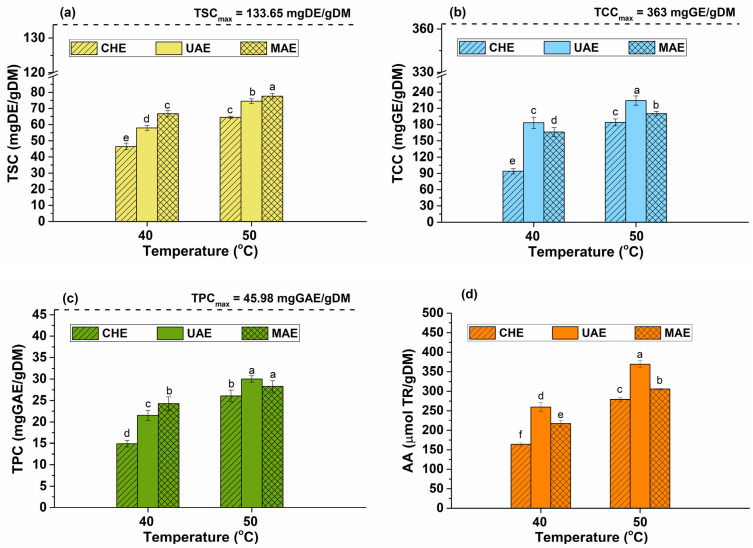
Influence of temperature on the total content of bioactive compounds: (**a**) TSC, (**b**) TCC, (**c**) TPC, and (**d**) AA—ivy leaves-to-solvent ratio of 1/20 (*w*/*v*), extraction time of 10 min, ethanol concentration in water of 80%, stirring rate of 900 rpm, ultrasound amplitude of 40%. The significant difference between groups (*p* < 0.05, ANOVA and Duncan’s post hoc *t*-tests) are highlighted by different letters (a–f).

**Figure 4 plants-12-03908-f004:**
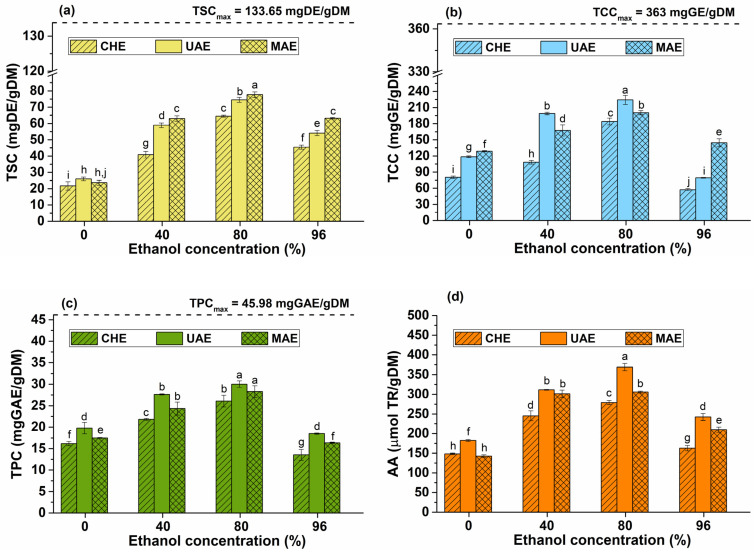
Influence of ethanol concentration on the total content of bioactive compounds: (**a**) TSC, (**b**) TCC, (**c**) TPC, and (**d**) AA—plant material to solvent ratio of 1/20 (*w*/*v*), extraction time of 10 min, temperature of 50 °C, stirring rate of 900 rpm, ultrasound amplitude of 40%. The significant difference between groups (*p* < 0.05, ANOVA and Duncan’s post hoc *t*-tests) are highlighted by different letters (a–j).

**Figure 5 plants-12-03908-f005:**
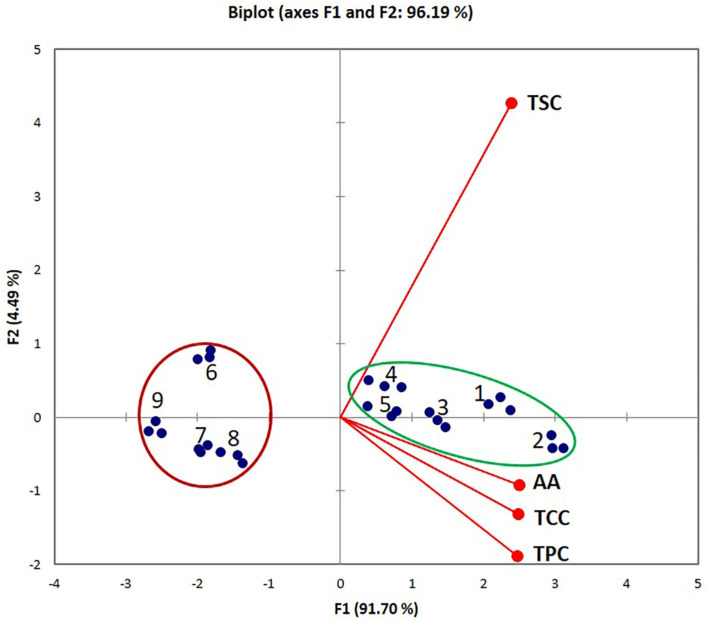
Projections of variables (TSC, TCC, TPC, and AA) and extraction methods (1−9) on the factor-plane PC1−PC2.

**Figure 6 plants-12-03908-f006:**
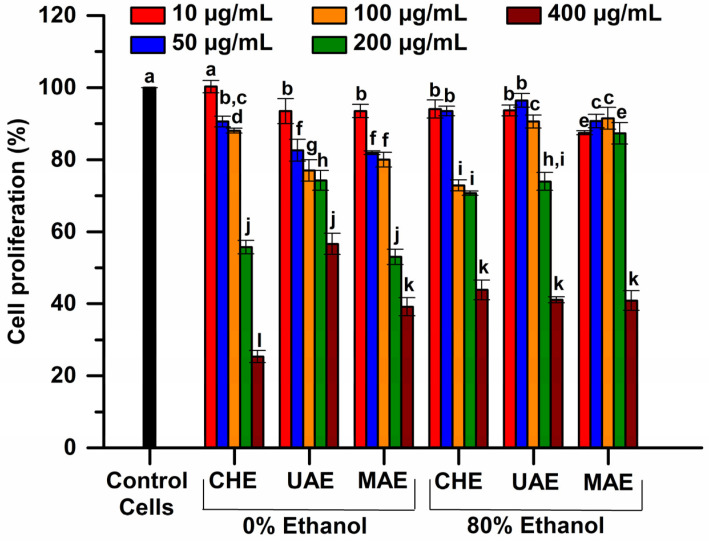
Cell proliferation of NCTC fibroblasts induced by the plant extracts at 48 h of treatment, determined via the MTT assay (extraction conditions of the selected extracts: temperature of 50 °C, extraction time of 10 min). The significant difference between groups (*p* < 0.05, ANOVA and Duncan’s post hoc *t*-tests) are highlighted by different letters (a–l).

**Figure 7 plants-12-03908-f007:**
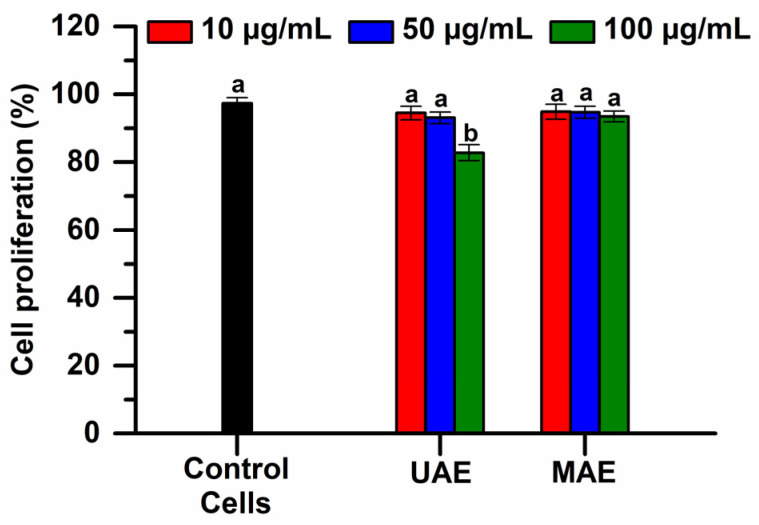
Cell viability of RAW 264.7 cells assessed by Live and Dead assay, after 48 h of treatment with the selected extracts (extraction conditions of the selected extracts: temperature of 50 °C, ethanol concentration in water of 80%, extraction time of 10 min). The significant difference between groups (*p* < 0.05, ANOVA and Duncan’s post hoc *t*-tests) are highlighted by different letters (a,b).

**Table 1 plants-12-03908-t001:** Energy consumption during phytocompound extraction (extraction time of 10 min).

Method Description	Total Energy (kJ)	Total Amount of Phytocompounds Extracted from Ivy Leaves (g/g DM)	Specific Energy (kJ/g of Phytocompounds)
CHE, 50 °C, 80% Ethanol	378	0.2746	1376.5
MAE, 50 °C, 80% Ethanol	9.0	0.3062	29.5
UAE, 50 °C, 80% Ethanol	18.0	0.3287	54.9
CHE, 40 °C, 80% Ethanol	360	0.1553	2318.8
MAE, 40 °C, 80% Ethanol	7.1	0.2613	27.2
UAE, 40 °C, 80% Ethanol	16.2	0.2627	61.6
CHE, 50 °C, 0% Ethanol	378	0.1182	3196.8
MAE, 50 °C, 0% Ethanol	9.2	0.1700	54.2
UAE, 50 °C, 0% Ethanol	28.0	0.1643	170.2
CHE, 50 °C, 40% Ethanol	378	0.1710	2210.1
MAE, 50 °C, 40% Ethanol	9.1	0.2548	35.7
UAE, 50 °C, 40% Ethanol	23.0	0.2853	80.6
CHE, 50 °C, 96% Ethanol	378	0.1162	3252.1
MAE, 50 °C, 96% Ethanol	8.2	0.2240	36.5
UAE, 50 °C, 96% Ethanol	17.4	0.1521	114.4

## Data Availability

Data are contained within the article and Appendix A.

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
