# Peer review of "Green Extraction Techniques of Phytochemicals from Hedera helix L. and In Vitro Characterization of the Extracts"

_plants, 2023, doi:10.3390/plants12223908_

Round 1
Reviewer 1 Report
Comments and Suggestions for Authors
Comments for authors:
1. Overall conclusion is missing in the Abstract section.
2. Line 56-77. The text should be reduced excluding well-known facts.
3. Abbreviations should be defined just the first time they are mentioned in the text.
4. The main drawback of the study is lack of identification and quantification of individual compounds and their relation with cytotoxicity, as well as other bioactivities.
5. Please, check the text for typos (italics, H. helix throughout the text...).
6. Subsection 2.2. More clear explanation of the methods used for the extraction of bioactive compounds. Were the same conditions used for each extraction (solvent, ratio, time)? Microwave power is missing.
7. Please, use term compounds instead of components.
8. Please, indicate practical application of the findings in the Conclusion section.
Author Response
Dear reviewer,
We very much appreciate the constructive and critical suggestions and comments, which have been very helpful in improving the quality of the manuscript. It has been revised in detail according to the comments, and all the comments were incorporated into the revised manuscript. The corrections have been marked up in the revised manuscript in red color. Our responses (in RED color) to all the comments are stated below:
- Overall conclusion is missing in the Abstract section.
Response: Thank you for your valuable comment. An overall conclusion was inserted in the Abstract section (please, see page 1, lines 28-30):
” Summarizing the above, both MAE and UAE can be employed as green and efficient methods to produce extracts rich in bioactive compounds with strong antioxidant properties and good non-cytotoxic activity.”
- Line 56-77. The text should be reduced excluding well-known facts.
Response: Thank you for the useful comment. The introduction was improved by reducing the well-known facts (please, see page 2, lines 63-74):
“The microwave assisted extraction (MAE) process consists of heating both the solvent and plant tissue, thus improving the kinetics of extraction. During the microwave heating the plant material is exposed to microwave energy, creating a temperature gradient that leads to a decrease in the chemical potential of the plant cell which will result in the cell wall rupture and, therefore, to an easy release of the targeted compounds [20-23]. Ultrasound assisted extraction (UAE) is based on the propagation of acoustic waves in liquids that produces the cavitation phenomenon. It is described by the generation of microbubbles or cavities, which collapse when they reach a maximum size. The presence of a solid material near the collapsing bubbles can produce an asymmetric collapse. Thus, the cavitation can promote rupture of the cell wall and, therefore, leading to an increase in mass transfer rate between solvent and plant material, also increasing the permeability of cell walls [24-27].”
- Abbreviations should be defined just the first time they are mentioned in the text.
Response: Thank you for the suggestion. According to the author’s guideline, the abbreviations should be defined the first time they appear in each of three sections: the abstract, the main text, the first figure or table. Thus, we defined the abbreviation in accordance with the journal guideline and your suggestion.
- The main drawback of the study is lack of identification and quantification of individual compounds and their relation with cytotoxicity, as well as other bioactivities.
Response: Thank you for the comment. The aim of this study was to establish the best extraction conditions for both non-conventional methods in accordance with different classes of bioactive compounds. The focus was on extraction techniques, rather than individual compounds found in ivy leaves or some specific individual compound within each phytochemicals class. This is why we reported the saponins, polyphenols, and carbohydrates in general, as a group, without identifying individual compounds. By this strategy, we accomplished to provide preliminary insights about non-conventional extraction techniques employed on ivy leaves. Subsequent studies, focusing on identifying compounds and optimizing the extraction of bioactive compounds, should be conducted to comprehensively evaluate non-conventional extractions and their overall potential.
- Please, check the text for typos (italics, H. helix throughout the text...).
Response: Thank you for noticing. The Latin name of plant (Hedera helix) was modified throughout the manuscript (in red color), in accordance with your suggestion.
- Subsection 2.2. More clear explanation of the methods used for the extraction of bioactive compounds. Were the same conditions used for each extraction (solvent, ratio, time)? Microwave power is missing.
Response: Thank you for your useful suggestions on the manuscript. For a better understanding, the following paragraph was inserted into the manuscript (please, see page 13, lines 408-413):
“All three methods (CHE, UAE, and MAE) were conducted under the following extraction conditions:
- mixtures of 0, 40, 80, and 96% ethanol in water as solvent;
- plant material to solvent ratio of 1/20 (w/v);
- temperatures of 40 and 50 °C;
- extraction times of 1, 5, 10, and 20 min.”
The following sentence about microwave power was also inserted into the 3.2 subsection (please, see page 13, lines 422-423):
“The microwave power was between 12 and 27 W depending on the extraction time, temperature, and ethanol concentration.”
- Please, use term compounds instead of components.
Response: Thank you for the suggestion. The term “components” was modified throughout the manuscript (in red color), in accordance with your suggestion.
- Please, indicate practical application of the findings in the Conclusion section.
Response: Thank you for the useful comment. The potential applications of ivy extracts were introduced into the Conclusion section. Thus, at page 16, lines 537-541 a new paragraph was introduced as follows:
“The resulted Hedera helix extracts could potentially find applications in pharmaceutical field as alternative sources of antioxidant, antimicrobial, and anti-inflammatory products. Overall, future studies focused on identification of the compounds and optimization of bioactive compound extraction should be performed for complete evaluation of non-conventional extractions and their overall potential.”
Reviewer 2 Report
Comments and Suggestions for Authors
The scientific literature contains many scientific publications that analyze extraction methods and their effects on obtaining biologically active compounds.
Why was ethanol chosen for use?
Line 2 should be changed - Method Description, shortened.
The presented list of literature is quite small, the latest literature should be added.
Comments on the Quality of English LanguageModerate editing of English language required.
Author Response
Dear reviewer,
We very much appreciate the constructive and critical suggestions and comments, which have been very helpful in improving the quality of the manuscript. It has been revised in detail according to the comments, and all the comments were incorporated into the revised manuscript. The corrections have been marked up in the revised manuscript in red color. Our responses (in RED color) to all the comments are stated below:
The scientific literature contains many scientific publications that analyze extraction methods and their effects on obtaining biologically active compounds.
- Why was ethanol chosen for use?
Response: Thank you for your useful comment. For a better understanding of the ethanol choice as solvent, the following paragraph was inserted into the manuscript (please, see page 8, line 238-249):
“For example, some solvents remain transparent to microwaves. This means that microwave energy can only be absorbed by dielectric solvents. The higher the dissipation factor of solvent, the better its microwave absorption [35]. Regarding ultrasounds, the most important properties are the viscosity, vapor pressure, and surface tension of solvent. High surface tension and low viscosity result in increased molecular interactions in the solvent, causing cavitation to be initiated at a higher energy threshold. The cavity bubbles are more violently collapsed in low vapor pressure solvents than in those with high vapor pressure [36].
One of the solvents that fulfills the above-mentioned characteristics and can be successfully used for both UAE and MAE is ethanol. Moreover, ethanol is a green solvent widely used for the extraction of bioactive compounds from different plants due to its low toxicity.”
- Line 2 should be changed - Method Description, shortened.
Response: Thank you for the useful comment. The introduction was improved by reducing the well-known facts (please, see page 2, lines 63-74):
“The microwave assisted extraction (MAE) process consists of heating both the solvent and plant tissue, thus improving the kinetics of extraction. During the microwave heating the plant material is exposed to microwave energy, creating a temperature gradient that leads to a decrease in the chemical potential of the plant cell which will result in the cell wall rupture and, therefore, to an easy release of the targeted compounds [20-23]. Ultrasound assisted extraction (UAE) is based on the propagation of acoustic waves in liquids that produces the cavitation phenomenon. It is described by the generation of microbubbles or cavities, which collapse when they reach a maximum size. The presence of a solid material near the collapsing bubbles can produce an asymmetric collapse. Thus, the cavitation can promote rupture of the cell wall and, therefore, leading to an increase in mass transfer rate between solvent and plant material, also increasing the permeability of cell walls [24-27].”
- The presented list of literature is quite small, the latest literature should be added.
Response: Thank you for the useful suggestion. More references and also the recent ones were introduced throughout the manuscript. See references:
- Zaiter, A.; Becker, L.; Baudelaire, E.; Dicko, A. Optimum polyphenol and triterpene contents of Hedera helix ( L. ) and Scrophularia nodosa ( L. ): The role of powder particle size. Microchem J 2018, 137, 168-173, doi:10.1016/j.microc.2017.10.011.
- Lopez-Avila, V.; Luque de Castro, M.D. Microwave-assisted extraction. In Elsevier Reference Module in Chemistry, Molecular Sciences and Chemical Engineering, Reedijk, J., Ed.; Elsevier Inc.: Waltham, MA, USA, 2014, pp. 1-17 doi:10.1016/b978-0-12-409547-2.11172-2.
- Gomez, L.; Tiwari, B.; Garcia-Vaquero, M. Emerging extraction techniques: Microwave-assisted extraction. In Sustainable Seaweed Technologies Cultivation, Biorefinery, and Applications, Torres, M. D.; Kraan, S.; Dominguez, H., Ed.; Elsevier Inc., 2020, pp. 207-224, doi:10.1016/b978-0-12-817943-7.00008-1.
- Kumar, K.; Srivastav, S.; Sharanagat, V.S. Ultrasound assisted extraction (UAE) of bioactive compounds from fruit and vegetable processing by-products: A review. Ultrason Sonochem 2021, 70, 105325, doi:10.1016/j.ultsonch.2020.105325.
- Fadimu, G.J.; Ghafoor, K.; Babiker, E.E.; Al-Juhaimi, F.; Abdulraheem, R.A.; Adenekan, M.K. Ultrasound-assisted process for optimal recovery of phenolic compounds from watermelon (Citrullus lanatus) seed and peel. J Food Meas Charact 2020, 14, 1784-1793, doi:10.1007/s11694-020-00426-z.
- Mason, T.J.; Vinatoru, M. Sonochemistry: Fundamentals and Evolution; Walter de Gruyter GmbH: Berlin/Boston, Germany/USA, 2023.
- Apak, R.; Guclu, K.; Ozyurek, M.; Karademir, S.E.; Altun, M. Total antioxidant capacity assay of human serum using copper(II)-neocuproine as chromogenic oxidant: the CUPRAC method. Free Radic Res 2005, 39, 949-961, doi:10.1080/10715760500210145.
- Christodoulou, M.C.; Orellana Palacios, J.C.; Hesami, G.; Jafarzadeh, S.; Lorenzo, J.M.; Domínguez, R.; Moreno, A.; Hadidi, M. Spectrophotometric methods for measurement of antioxidant activity in food and pharmaceuticals. Antioxidants 2022, 11, 2213, doi:10.3390/antiox11112213.
- Mandal, V.; Mohan, Y.; Hemalatha, S. Microwave assisted extraction-An innovative and promising extraction tool for medicinal plant research. Pharmacogn Rev 2007, 1, 7-18.
- Vernès, L.; Vian, M.; Chemat, F. Ultrasound and microwave as green tools for solid-liquid extraction. In Liquid-Phase Extraction, Poole, C. F., Ed.; Elsevier Inc., 2020, pp. 355-374, doi:10.1016/b978-0-12-816911-7.00012-8.
Reviewer 3 Report
Comments and Suggestions for Authors
Dear Dr. Popa
After carefully reviewing this manuscript, I have gained a lot. I think this is a very meaningful study and I express my sincere congratulations! But I have two suggestions about your manuscripts and one question that you need to modify and answer.
The first suggestion: Please replace the data in Table 1 with a method of presentation. I recommend using the columnar diagram. Because in my opinion, when readers read your papers, there is no way to find the best extraction method the first time. The expression of this columnar diagram can be similar to Fig1 or Fig2 in your manuscript.
Second suggestion: You can ignore this suggestion! I recommend deleting the " vitro Characterization" section. In my opinion, "Plants" is a journal that focuses on plant science,but this part of the content is toxicology. If you are really interested in toxicology, I suggest that you do in-depth research and refine this part of the content, and I welcome you to submit a focused manuscript to MDPI Press again.
Question: I can't understand the content of the word "antioxidant" in your manuscript. In my opinion, a variety of constituents in medicinal plants have antioxidant effects, such as the TSC and TCC mentioned in your papers. Or is it ascorbic acid (ASA) or glutathione (GSH), will the material processing and extraction methods you mention cause these ingredients? Therefore, I suggest you clarify which type of compound or specific individual ingredient.
Author Response
Dear reviewer,
We very much appreciate the constructive and critical suggestions and comments, which have been very helpful in improving the quality of the manuscript. It has been revised in detail according to the comments, and all the comments were incorporated into the revised manuscript. The corrections have been marked up in the revised manuscript in red color. Our responses (in RED color) to all the comments are stated below:
After carefully reviewing this manuscript, I have gained a lot. I think this is a very meaningful study and I express my sincere congratulations! But I have two suggestions about your manuscripts and one question that you need to modify and answer.
The first suggestion: Please replace the data in Table 1 with a method of presentation. I recommend using the columnar diagram. Because in my opinion, when readers read your papers, there is no way to find the best extraction method the first time. The expression of this columnar diagram can be similar to Fig1 or Fig2 in your manuscript.
Response: Thank you for the useful suggestion. The data presented in Table 1 were represented as a columnar diagram in accordance with your suggestion. Please, see Figure 1 on page 4, line 125-130.
Second suggestion: You can ignore this suggestion! I recommend deleting the " vitro Characterization" section. In my opinion, "Plants" is a journal that focuses on plant science,but this part of the content is toxicology. If you are really interested in toxicology, I suggest that you do in-depth research and refine this part of the content, and I welcome you to submit a focused manuscript to MDPI Press again.
Response: Thank you for the suggestion. The aim of this work was to highlight the applicability of the extracts obtained from inedible plants (ivy leaves) with beneficial properties on human health as alternative sources of antioxidant, antimicrobial, and anti-inflammatory products and other applications in foods, agriculture, cosmetics etc.. The biological activity was added to show once more the applicability of the extracts. Anyway, future studies focused on identification of the compounds and in-depth research of toxicology for complete evaluation of non-conventional extractions and overall potential of Hedera helix should be performed.
Question: I can't understand the content of the word "antioxidant" in your manuscript. In my opinion, a variety of constituents in medicinal plants have antioxidant effects, such as the TSC and TCC mentioned in your papers. Or is it ascorbic acid (ASA) or glutathione (GSH), will the material processing and extraction methods you mention cause these ingredients? Therefore, I suggest you clarify which type of compound or specific individual ingredient.
Response: Thank you for the useful comment. The Results and Discussion section was improved by introducing the following paragraph about CUPRAC analysis (please, see page 3, lines 112-124):
“The total antioxidant potential of the samples was determined by cupric reducing antioxidant power (CUPRAC) colorimetric assay. Cu²⁺ reduction is a common method for assessing electron donation activity and is an important mechanism of antioxidants. Thus, an investigation into the Hedera helix extracts' capacity to reduce Cu(II) in order to evaluate their electron-donating abilities was conducted. High values of AA suggest a high reducing activity. Using copper(II)-neocuproine reagent as the chromogenic oxidant, the CUPRAC assay is based on the reduction of Cu(II) to Cu(I) by antioxidants found in plant extracts. Due to their high concentration of saponins, polyphenols, and carbohydrates, which function as electron donors, the extracts may have an antioxidant mechanism that contributes to their cupric reducing ability [30,31]. The AA of the extracts obtained by multiple successive extractions, for each method, is also shown in Figure 1. The highest AA values were obtained for UAE (368.98±9.01 µmol TR/g DM), followed by MAE (305.70±2.36 µmol TR/g DM).”
Also, a common method for assessing the connection between phytochemicals and antioxidant activity in plants is correlation analysis. According to the study's findings, there is a statistically positively significant correlation between saponins, carbohydrates, and polyphenols and antioxidant properties (as it is described at page 10 lines 312-316).
Reviewer 4 Report
Comments and Suggestions for Authors
This work is well written and comprehensive. Nevertheless, in my opinion, some improvements could be made:
In vitro should be in Italic type.
The first sentence of the Abstract needs to be corrected (row 15). Phytochemicals could not lead to extracts.
In Introduction section, there are several large parts of the text (rows 60-77) where only 1-2 authors are cited. I suggest more authors to be cited in this text. In my opinion, the last part of Introduction (rows 78-90), was done in the opposite order. First should be placed the works of the topic (the investigation gap), and then the following aim of the study.
In Results and Discussion, I think it is good to compare the results received in this study to the results of other authors working on the same topic, if such data exist. For example, TPC for UAE is 28.3 GAE/gDM. It could be compared with such data received by other authors for the same extraction method. Also, such comparisons with other methods, described in row 57, could be made so the readers can understand the efficacy of the green methods compared to the other aforementioned methods.
In Materials and Methods, you should put the name of the country after the company from which you purchased the products.
Author Response
Dear reviewer,
We very much appreciate the constructive and critical suggestions and comments, which have been very helpful in improving the quality of the manuscript. It has been revised in detail according to the comments, and all the comments were incorporated into the revised manuscript. The corrections have been marked up in the revised manuscript in red color. Our responses (in RED color) to all the comments are stated below:
This work is well written and comprehensive. Nevertheless, in my opinion, some improvements could be made:
- In vitro should be in Italic type.
Response: Thank you for your useful suggestion. The word ”in vitro” was modified throughout the manuscript, in accordance with your suggestion.
- The first sentence of the Abstract needs to be corrected (row 15). Phytochemicals could not lead to extracts.
Response: Thank you for the valuable comment. The first sentence of the Abstract was modified as follows (please, see page 1, lines 15-16):
“Hedera helix L. contains phytochemicals with good biological properties which are beneficial to human health and can be used to protect plants against different diseases.”
- In Introduction section, there are several large parts of the text (rows 60-77) where only 1-2 authors are cited. I suggest more authors to be cited in this text. In my opinion, the last part of Introduction (rows 78-90), was done in the opposite order. First should be placed the works of the topic (the investigation gap), and then the following aim of the study.
Response: Thank you for the useful comments. The introduction was improved by reducing the well-known facts and several additional references were inserted (please, see page 2, lines 63-74):
“The microwave assisted extraction (MAE) process consists of heating both the solvent and plant tissue, thus improving the kinetics of extraction. During the microwave heating the plant material is exposed to microwave energy, creating a temperature gradient that leads to a decrease in the chemical potential of the plant cell which will result in the cell wall rupture and, therefore, to an easy release of the targeted compounds [20-23]. Ultrasound assisted extraction (UAE) is based on the propagation of acoustic waves in liquids that produces the cavitation phenomenon. It is described by the generation of microbubbles or cavities, which collapse when they reach a maximum size. The presence of a solid material near the collapsing bubbles can produce an asymmetric collapse. Thus, the cavitation can promote rupture of the cell wall and, therefore, leading to an increase in mass transfer rate between solvent and plant material, also increasing the permeability of cell walls [24-27].”
The last part of the introduction was also modified in accordance with your suggestion. Thus, the paragraph on page 2, rows 75-88 was changed as follows:
“The evaluation and screening of the most suitable extraction method and the most favorable parameters for extracting various bioactive compounds from Hedera helix leaves are reported in this study. To the best of our knowledge, a comparison of these phytochemicals’ extraction methods has not been reported yet and the microwave assisted extraction of different bioactive compounds from Hedera helix leaves has been employed, for the first time, in this work. There is only one study which uses microwaves, but the researchers applied a microwave pre-treatment of the Caulis hederae sinensis leaves, the extraction of saponins being performed with supercritical fluids [28]. The aim of this study was to evaluate the Hedera helix leaves extracts as potential sources of bioactive compounds, such as saponins, carbohydrates, and polyphenols, with therapeutic use. The antioxidant activity of the extracts is also reported. Finally, in vitro analysis of the extracts richest in bioactive compounds were conducted in order to evaluate their cytotoxicity on NCTC normal fibroblast cells and the influence on the DNA content of RAW 264.7 murine macrophages. “
- In Results and Discussion, I think it is good to compare the results received in this study to the results of other authors working on the same topic, if such data exist. For example, TPC for UAE is 28.3 GAE/gDM. It could be compared with such data received by other authors for the same extraction method. Also, such comparisons with other methods, described in row 57, could be made so the readers can understand the efficacy of the green methods compared to the other aforementioned methods.
Response: Thank you for the valuable comments. Two comparisons were introduced into the Results and Discussion section. Thus, at page 6, lines 190-192 a new phrase was inserted as follows:
“The maximum TPC obtained by non-conventional methods is comparable with that obtained by maceration in 12 h – 28.3 mg GAE/g reported by Zaiter et al [30]. However, in this study by MAE and UAE, the content was achieved in a much shorter time (10 min).”
And at page 7, lines 229-233 a new phrase was introduced as follows:
“These results are better than those obtained by conventional methods reported in the literature. By heat reflux extraction, Tatia et al achieved values of 30.02 mg DE/g and 199.27 mmol TR/g for TSC and AA, respectively [17], which are lower than those obtained in this study by UAE and MAE.”
- In Materials and Methods, you should put the name of the country after the company from which you purchased the products.
Response: Thank you for your suggestion. The name of the country was modified in accordance with your suggestion (please, see page 13).
Reviewer 5 Report
Comments and Suggestions for Authors
Dear Authots, I have read the MS entitled Green Extraction Techniques of Phytochemicals from Hedera helix L and in vitro Characterization of the Extracts. I have some observations.
-please check and correct all over text: "we" should be replaced with other terms, such as "the authors".
-always use passive voice, not active voice
-please check and correct the numbers for materials and methods and results. There is a misprint there...
-please check that figures and their title are on the same page
-please check experimental protocol, as control of the vegetal material seems to be missing.
Comments on the Quality of English LanguageEnglish language is ok, small changes should be made from "we" to other terms and from active voice to passive voice.
Author Response
Dear reviewer,
We very much appreciate the constructive and critical suggestions and comments, which have been very helpful in improving the quality of the manuscript. It has been revised in detail according to the comments, and all the comments were incorporated into the revised manuscript. The corrections have been marked up in the revised manuscript in red color. Our responses (in RED color) to all the comments are stated below:
Dear Authots, I have read the MS entitled Green Extraction Techniques of Phytochemicals from Hedera helix L and in vitro Characterization of the Extracts. I have some observations.
-please check and correct all over text: "we" should be replaced with other terms, such as "the authors".
-always use passive voice, not active voice
Response: Thank you for your useful comments. In accordance with your suggestion, the passive voice was used, and the phrases were modified throughout the manuscript.
-please check and correct the numbers for materials and methods and results. There is a misprint there...
Response: Thank you for noticing. The number of sections were modified throughout the manuscript.
-please check that figures and their title are on the same page
Response: Thank you for your comment. The figures and their title were placed on the same page.
-please check experimental protocol, as control of the vegetal material seems to be missing.
Response: Thank you for the useful comment. For a better understanding, the description procedure of the control experiment was modified as follows (please, see page 13-14, lines 425-429):
“The CHEs were performed in order to compare the efficiency of non-conventional extraction methods (UAE and MAE) of bioactive compounds from ivy leaves. The experiments under CHE were carried out in the same conditions as for non-conventional techniques using the jacketed glass reactor as for UAE, but without applying the ultrasounds, and using a stirring rate of 900 rpm.”
Round 2
Reviewer 1 Report
Comments and Suggestions for Authors
The authors significantly improved the quality of the manuscript.
Author Response
Dear reviewer,
Thank you for your comments on the manuscript and for appreciating our work.
Reviewer 2 Report
Comments and Suggestions for Authors
Dear Authors,
Thank you and corections for publication.
Comments on the Quality of English LanguageMinor editing of English language required.
Author Response
Dear reviewer,
We very much appreciate the constructive and critical suggestions and comments, which have been very helpful in improving the quality of the manuscript.
We have revised and refined the manuscript in accordance with your suggestion. The language changes have been highlighted in blue color within the manuscript.
Hopefully, we have addressed all your concerns.
Reviewer 5 Report
Comments and Suggestions for Authors
Dear Authors ,thank you for incorporating most of my observations into your article. However, I still find there are some gaps in presenting the experimental protocol regarding used vegetal material, re. ivy leaves. What leaves were harvested? When? How were they harvested? How many? Were there more variants regarding vegetal material or just regarding extraction method?
Please add more reproductible data regarding the vegetal material taken into analysis.
Author Response
Dear reviewer,
We very much appreciate the constructive and critical suggestions and comments, which have been very helpful in improving the quality of the manuscript. It has been revised in detail according to the comments, and all the comments were incorporated into the revised manuscript. The corrections have been marked up in the revised manuscript in blue color. Our responses (in RED color) to all the comments are stated below:
Dear Authors ,thank you for incorporating most of my observations into your article. However, I still find there are some gaps in presenting the experimental protocol regarding used vegetal material, re. ivy leaves. What leaves were harvested? When? How were they harvested? How many? Were there more variants regarding vegetal material or just regarding extraction method?
Please add more reproductible data regarding the vegetal material taken into analysis.
Response: Thank you for the comments. For a better understanding, the following paragraph (in blue color) was introduced into the manuscript (please, see page 13, lines 382-386):
“The leaves were harvested manually, without stems. To ensure the reproducibility of the extraction methods, leaves from the same lot were used. Information about the ivy leaves used in this study can be obtained from the voucher plant specimen number 407754, which is archived at the Botanical Garden in Bucharest, Romania.”
Hopefully, we have addressed all your concerns.